# Evaluation of Dynamics, Demography and Estimation of Free-Roaming Dog Population in Herat City, Afghanistan

**DOI:** 10.3390/ani13071126

**Published:** 2023-03-23

**Authors:** Zabihullah Nasiry, Mazlina Mazlan, Mustapha M. Noordin, Mohd Azmi Mohd Lila

**Affiliations:** 1Department of Veterinary Pathology & Microbiology, Faculty of Veterinary Medicine, Universiti Putra Malaysia (UPM), Serdang 43400, Selangor, Malaysia; 2Department of Paraclinic, Faculty of Veterinary Science, Herat University, Herat 3001, Afghanistan

**Keywords:** free-roaming dogs, demography, rabies, photographic sight–resight method, Herat city

## Abstract

**Simple Summary:**

Understanding the demography, population size, and dynamics of free-roaming dogs (FRD) is essential for developing an effective dog-population-management and rabies-control programme. The existence of an uncontrolled FRD population in Herat, Afghanistan, is causing threats to public health and has raised concerns for their welfare. The aim of this study was to evaluate the demographic structures and estimate the FRD population size in Herat city, Afghanistan. The photographic sight–resight method utilized and human-to-FRD ratio in this study yielded a density of 10 dogs/km^2^ and an estimated ratio of 315:1. Dogs were predominantly adult males with a good health status and ideal body-condition score. The knowledge of FRD demography and population size provided by this study could be used by the local government for much more effective planning, implementation and monitoring of dog-population-control programmes.

**Abstract:**

FRDs pose a serious challenge in countries where dog-bite-related rabies is endemic. Understanding the size and core demographic characteristics of FRD populations is essential for the planning and implementation of effective dog-population and canine-rabies-control programmes. The photographic sight–resight method was used to estimate the FRD population and evaluate its demographic characteristics in Herat city. A total of 928 free-roaming dogs (FRD) were identified through 3172 sightings, and the total free-roaming population was estimated to amount to 1821 (95% CI: 1565–2077), which led to the estimation of 10 dogs/km^2^ and the human-to-FRD ratio of 315:1. The male-to-female ratio was 2.85:1. The majority of them were healthy, with an ideal body score. Although the FRD density is considered low, it is still a concern and significant, since the majority of the people are unaware of the importance of canine populations in the transmission of zoonotic diseases such as rabies, and there were no specific measures for managing and controlling FRD populations. The information gained can be useful in animal health planning to design effective dog-population-control programmes, and for the planning of national rabies-prevention programmes.

## 1. Introduction

The familiar, widely distributed and phenotypically diversified domestic dog (*Canis lupus familiaris*) is one of our closest animal companions [1,2,3,4]. It is the most popular owned pet species in the world [5,6]. Dogs are considered part of the family, particularly in developed countries [7]. Although pet dogs are usually restricted in their behaviour, a significant proportion of global dog population are free-roaming on the streets (and are commonly known as free-roaming dogs/FRDs) [8]. Some reports have estimated that 75% of the 700 million world dogs population are stray FRDs [9,10,11,12]. Other reports estimated that there were 800 million dogs in the world and that 300 million were FRDs [13], explaining why they are a common sight in many countries [14]. The management of FRDs is challenging (because of relatively high fecundity) and is a significant problem in both developing and developed countries [15].

FRDs pose serious challenges in a community, such as risks of attacks or bites, the transmission of diseases, threats to wildlife population, vehicular accidents and pollution [8]. One of the great problems of FRDs is the threat of rabies, as the epidemiology of human rabies is closely connected with that of dogs, which are responsible for almost 99% of global human rabies deaths [12,16,17,18,19,20]. This has led to WHO recommending the control of rabies in FRDs through annual mass vaccination of FRDs, with coverage of at least 70% of the population [21].

Canine rabies is a zoonotic, fatal and progressive neurological infection caused by Lyssaviruses of the family Rhabdoviridae [22]. The disease causes an estimated 59,000 human deaths globally every year and an estimated loss of 3.7 million in disability-adjusted life years (DALYs) annually [18,19,23,24,25,26,27,28,29], which is more than any other zoonotic disease [16,19]. The disease is almost 100% lethal once clinical signs emerge [16,30]. Human rabies is transmitted from bites by infected animals, predominantly dogs [28,31]. It is reported that 99% of global rabies deaths are attributed to the transmission of rabies through dog bites [18]. The threat of this deadly disease affects more than 3.3 billion people globally [24], with the world’s poorest regions at highest risk [28]. The rabies virus is endemic, and dog bites are common in countries where over half of world’s population reside at [19] It was estimated that more than 29 million people worldwide receive rabies postexposure prophylaxis each year for dog bites [19,23]. Rabies accounts for economic losses of about USD 8.6 billion annually, mainly due to premature deaths and postexposure treatment (PET) costs [28]. Almost all mammal species are susceptible to rabies, and it has encroached livestock yields, with an estimated loss of USD 12.3 billion in Africa and Asia [24]. The disease can be prevented and controlled through the mass vaccination of dogs, movement restrictions of dogs and quarantining imported dogs [32,33]. Human rabies is almost entirely preventable through the prompt administration of postexposure prophylaxis (PEP) to bite victims [34].

Rabies is endemic in Afghanistan, with hundreds of deaths annually [35]. The country has the highest number of human rabies cases in South Asia, at 5.7 per 100,000 people [34].The likely root problems of the ongoing rabies being endemic in Afghanistan are attributed to inaccurate epidemiological information and the absence of specific and sustainable rabies-prevention and -control measures [35]. The country started to adopt mass dog vaccination with a neutering programme in Kabul, and tens of thousands of dogs were vaccinated against rabies between 2017 and 2019. However, the abundant number of FRDs, lack of regular vaccination, absence of regular-animal-birth-control programmes, limited human resources, and insufficient funds are the significant constraints in controlling rabies [34].

Rabies epidemiology is closely associated with dog ecology [36,37]. Additionally, it is now generally believed that a good knowledge of local domestic dog ecology is essential for implementing an effective rabies-control strategy [38]. Thus, understanding dog demographic characteristics and population density is essential for planning and monitoring dog-population management and control, as well as for designing appropriate rabies-control measures in dog populations [8,10,20,21,24,37,39]. Accordingly, many studies have been conducted worldwide on free-roaming dogs [7,14,17,20,21,37,39,40,41,42,43,44,45,46,47,48,49,50,51,52,53,54,55]. 

Herat is situated in the southwestern part of Afghanistan and it is the second most populated city in the country. The lack of epidemiological data on canine populations, coupled with ineffective control measures, means that canine rabies remains endemic in this province. Additionally, the existence of an uncontrolled FRD population has negatively influenced public health, leading to socioeconomic, political and animal welfare problems. The information generated by this research can be useful in animal health planning to design effective dog-population-management and -control programmes and can be valuable to the local government in planning national rabies-prevention programmes.

## 2. Materials and Methods

### 2.1. Study Area

The study was carried out in the Herat province (34.1769° N, 61.7006° E), which encompasses an area of 55,868.5 km^2^ and is situated in the southwestern part of Afghanistan, bordering Iran (Islam Qala Crossing) and Turkmenistan (Torghundi Crossing). It has internal borders with Badghis province in the north, Ghor in the east and Farah in the south. Herat, the provincial capital of Herat Province, is the densest and the second most populated city in the country, with more than 2 million people. Herat city covers 182 km^2^ and divided into 15 districts (Figure 1) (Afghanistan Statistical Yearbook, 2020). Herat city is quite densely populated, owing to being well known for its cultural heritage, industrial nature and manufacturing industries. 

### 2.2. Study Design and Sampling

The study was designed as a cross-sectional study and was conducted between September 2020 and April 2021 in all 15 districts of Herat city using photographic sight–resight methods to estimate and evaluate the FRD population. In this study, FRDs were defined as dogs that were not under the direct control of any person or were possibly owned but roaming during the survey period. The FRD survey was enhanced by dividing Herat city into 31 blocks, and each block (demarcated by large roads) covered an area of ~5 km^2^. Each block was visited twice daily, in the early morning and evening. During the field survey, representatives from each predetermined area were informed the day before a survey of the area was conducted.

### 2.3. Survey Methods

The survey was conducted using the photographic sight–resight (PSR) method. A preliminary PSR survey revealed that the most active times for FRDs were the early morning (4:30–6:30 a.m.) and evening (6:30–8:30 p.m.). Streets and alleys were traversed by a car at a maximum speed of 20 km/h. Once a dog or group of dogs were observed, photographs were taken by using a Nikon COOLPIX P510 digital camera (Nikon company, Tokyo, Japan) at appropriate angles from a convenient distance. All observable characteristics of each dog were recorded including its sex (male; female; not verified), age (puppy (<6 months); young (6 months to 1 year); adult (>1 year)), reproductive status (pregnant; lactating; female in oestrus; not verified), body-condition scoring, observable health status (lameness; emaciation; skin lesions or dermatitis (abnormal changes in any area of the skin); mad dog signs), details of coat colour (composition; primary or secondary colour), coat condition (wounded; healthy; clean; dirty) and coat hair condition (long; medium; short), along with its GPS location (using a Garmin 60CS device). In addition, the activity of the FRD at the time of the survey (active; inactive; feeding or foraging), its social behaviour (alone; in a group), its reaction during the surveyor (friendly approaches; low-posture approaches; growls or barks; attacks the surveyor; ignores the surveyors; runs away), its location (road; sidewalk; playground; field; parking lot; underneath a car; bushes; dump sites; slaughterhouse; biological-waste-disposal sites; other places) and its proximity (presence ≤ 20 m) to garbage-disposal sites, such as garbage dumps/points or temporary accumulated litter along the streets and alley, were recorded. 

### 2.4. Data Collection

All of the above parameters were provided in a predefined format to facilitate the data-collection process during the survey. Then, the photographs were reviewed to identify each dog by using distinguishable natural marks (on their flanks, ears, head and tail), body condition, details of coat colour, coat hair condition, age, sex, reproductive status and ears/tail amputation. The first time an individual dog was observed, this was defined as the initial sighting (capture) event. The subsequent encounters with the same dog were considered resighting (i.e., recapture). The identification of dogs relied on direct observation and photographs and, thus, none of the dogs were physically captured in this study. Following completion of the identification, each dog was given a unique identification code. The GPS location of each dog was loaded into GIS software (ArcMap version 10.4) and displayed on the map. Data through field survey using the PSR method were collected, then processed and analysed (Figure 2).

### 2.5. Data Management and Analysis

Collected data were computed using Microsoft Excel (version: 16.0.4266.1001). Descriptive statistics were produced and 95% confidence intervals were calculated by using GraphPad Prism version 8.0.2 for Windows (GraphPad software, Lajolla, CA, USA). The chi-square test was utilized to examine variation in different categories (sex, age group, body-condition score, reproductive status) between groups using GraphPad Prism. GPS points of FRDs were loaded into ArcGIS 10.4 to be displayed on the map. The estimation of the FRD populations was made using the Lincoln–Petersen formula with Chapman’s correction.
$$N=\left[\frac{\left(\;n1+1\right)\times \left(n2+1\right)}{\left(m2+1\right)}\right]-1$$

The variance of N was estimated using Seber’s formula [48]:$$var\left(N\right)=[\frac{\left(n1+1\right)\left(n2+1\right)\left(n1-m\right)\left(n2-m\right)}{{\left(m+1\right)}^{2}\left(m+1\right)}$$

An approximate 95% confidence interval for N was estimated as:$$N\;\pm 1.965\sqrt{var\left(N\right)}$$
where N is the total population size, n1 is the number of dogs sighted/photographed on the first occasion, n2 is the number of dogs sighted/photographed on the second occasion, and m2 is the number of dogs resighted.

## 3. Results

### 3.1. Number of FRDs

A total of 928 unique FRDs (predominantly male) were identified through 3172 reliable sightings in fifteen districts of Herat city during the survey period (Figure 3).

Photos were available for 99.1% of these dogs (due to restrictions on photography on the roads around security areas, political representations of foreign countries and poor quality of photographs). The total FRD population was estimated to be 1821 (95% CI: 1565–2077), leading to an estimate of 10 dogs/km^2^, with a human-to-dog ratio of 315:1 (3.17 dogs per 1000 people). Most dogs were identified during the morning shift (probably due to ample amount of uncollected garbage, and less human interference). The number of dogs identified in each district ranged from 27 to 123, bearing a linear relationship of R² = 0.0627, y = 1.8643x + 46.952. Most dogs were recorded in districts 5, 9 and 15, while the lowest FRD population density was recorded in districts 2, 10, 1 and 7.

### 3.2. Demography Characteristics

Out of the 928 FRDs, 60.6% (562/928; 95% CI = 65.6–57.6%) were males, 21.2% (197/928; 95% CI = 23.1–18.3%) females and 18.2% (169/928; 95% CI = 20–17.6%) were of unverifiable sex. Overall, the male dogs were significantly more represented than other categories (χ2 = 65.33, df = 28, *p* < 0.001). The male-to-female ratio was 2.85:1 (95% CI = 3.6:1–2.7:1). The variations in sex ratio between districts was 1.33–5.83:1. The majority of dogs, 85.5% (793/928; 95% CI = 92.7–81.4%), were observed to be more than 1 year old (adult), 10.3% (96/928; 95% CI = 11.4–10%) were ≥ 6 months (young) and 4.2% (39/928; 95% CI = 4.4–4%) were <6 months of age (puppy). Adults were the largest percentage of all the categories (χ2 = 57.63, df = 28, *p* < 0.001). Out of 197 identified female dogs, 9.13% (18/197; 95% CI = 10–8.8%) were lactating (identified based on visual observation/presence of infant puppies), 7.61% (15/197; 95% CI = 8.4–7.4%) were pregnant dogs (recognized based on body condition and behaviour) and 8.62% (17/197; 95% CI = 9.5–8.4%) of the dogs were in oestrus (recognized based on being followed by male dogs). No significant variation was recorded in the reproductive status compositions of FRDs in districts during the survey period (χ2 = 51.81, df = 39, *p* < 0.08). All demographic details (sex, age distribution and reproductive status) of these FRDs are displayed in Table 1.

### 3.3. Health Status and Body-Condition Score

Generally, the health status of FRDs was good, with 82% (764/928; 95% CI = 89.6–78.6%) being healthy, while lameness, excessive emaciation and skin lesions or dermatitis were recorded in 18% (164/928; 95% CI = 19–16.7%) of FRDs (Figure 4 and Figure 5). Based on coat hair condition, 89.98% (835/928; 95% CI = 97.7–85.8%) and 10.02% (93/928; 95% CI = 10.8–9.5%) of FRDs were assessed as healthy and wounded, respectively. Coat sanitary condition was recorded to be 74.9% (695/928; 95% CI = 81.4–71.5%) clean, 2.7% (25/928; 95% CI = 2.9–2.5%) shiny and 22.4% (208/928; 95% CI = 24.25–21.2%) dirty. 

The most common body-condition score (BCS) seen was BCS3, in around 60.13% (558/928; 95% CI = 65.6–57.6%) of the dogs, which is considered an ideal body condition (Figure 6). The amputation of the tail and ears of dogs is commonly carried out by dog owners when they want to keep the dogs as pets or guard dogs. We recorded ears/tails amputated in 33.65% (312/928; 95% CI = 36.6–32.2%) of free-roaming dogs, which might suggest that these particular dogs were owned abandoned dogs or owned free-roaming dogs.

### 3.4. Location of Free-Roaming Dogs during the Time of Survey

The location of dogs varied significantly (χ2 = 957.7, df = 154, *p* < 0.0001), and most dogs were seen on the roads (42.13%; 391/928; 95% CI = 45.8–40.3%) and sidewalks (32.76%; 304/928; 95% CI = 35.7–31.5%) (Table 2). The remaining FRDs were seen at dump sites (6.25%; 58/928; 7–6.24%), fields (6.03%; 56/928; 95% CI = 6.7–5.7%), boulevards (5.71%; 53/928; 95% CI = 6.4–5.6%), parking lots (2,59%; 24/928; 95% CI = 2.9–2.6%), slaughterhouses (2.37%; 22/928; 95% CI = 2.8–2.5%), parks (1.4%; 13/928; 95% CI = 1.6–1.43%), playgrounds, underneath cars and bushes (0.66%; 6/928; 95% CI = 0.68–0.65%) and at biological-waste-disposals sites (0.1%; 2/928; 95% CI = 0.12–0.11%). 

### 3.5. Reaction of Free-Roaming Dogs during the Time of Survey

A significant variation was observed in FRD reaction (χ2 = 278.9, df = 70, *p* < *0*.0001). Around 43.75% (406/928; 95% CI = 47.6–41.8%) of dogs did not tolerate human presence and fled whenever approached, which contrasts with 30.5% (283/928; 95% CI = 33.5–28.5%) of dogs, which had neutral approaches or ignored the surveyor during the survey. Slow approaches, growls and barks, attacks on the surveyor and friendly approaches were seen and recorded, respectively, in about 18.97% (176/928; 95% CI = 20.9–18.4%), 4.96% (46/728; 95% CI = 5.3–4.7%), 1.07% (10/928; 95% CI = 1.1–1%) and 0.75% (7/928; 95% CI = 0.8–0.74%) of dogs (Table 3).

### 3.6. Social Behaviour and Activity

Dogs were seen in groups significantly more often (68%, 95% CI = 73.4–64.5%) than alone (χ2 = 45.10, df = 14, *p* < 0.0001). Additionally, 68% (631/928; 95% CI = 73.9–46.9%) of them were active (exploring, walking, interacting, feeding and foraging) during the survey time, and 32% (297/928; 95% CI = 34.7–30.4%) inactive (standing, sitting, laying down and sleeping) (Table 4).

Out of 928 FRDs, 441 (48%; 95% CI = 51.5–45%) were seen ≤20 m from the garbage sites (including places which accumulations of garbage are left temporarily, in small amounts such as the roadsides and sidewalks and official dump sites) (Table 4 and Figure 7). Although no significant variation was observed in the activities of FRDs during the survey time (χ2 = 38.70, df = 28, *p* < 0.08), a significant difference was found in the number of dogs sighted within 20 m of garbage dumps/accumulated litter (χ2 = 127.9, df = 14, *p* < 0.0001) across the districts.

Out of the 629 (68%) dogs that were seen in packs, 60.2% were male dogs, 21.5% were female dogs and 18.3% were dogs with unverified gender. Out of the total male dogs, female dogs and dogs with unverified gender, 67% (379/562), 68.5% (135/197) and 68% (115/169) were seen in packs during the survey period, respectively. The most common composition of social structure was observed as MM/0, MM/UF, MM/MF and UM/UF (Figure 8). The social behaviour was varied in each district (Figure 9).

## 4. Discussion

Estimating and evaluating of FRD populations is critical for the planning and implementation of effective dog-population- and canine-rabies-control programmes, particularly in countries where rabies is endemic such as Afghanistan. Although many studies have been conducted elsewhere [7,14,21,37,39,42,44,47,48,49,50,51,52,53,54,55], this was the first study that evaluated and estimated the size of FRD populations in Herat city, Afghanistan.

As in many previous studies around the world [7,21,40,46,50,55,56,57], the photographic sight–resight method was utilized in this study to evaluate demographic characteristics and estimate the FRD population size in Herat city. This method has several advantages, such as reducing the cost of the survey, reducing the risk of researchers’ exposure to dog bites or rabies and reducing the scaring of targeted FRDs in the study population, which would result in them fleeing away (because individual dogs are not physically captured in this method and only photographed from a distance without disturbing their natural behaviour). In addition, the use of digital photography is useful during field surveys to reduce misidentification of the individual dogs, which can lead to inaccurate evaluation and estimation. Like many previous studies conducted elsewhere [47,48,58,59], the Lincoln–Petersen formula with Chapman’s correction was applied to estimate the FRD population. This method has an advantage over other methods as it requires only two sighting sessions [60].

The FRD per km^2^ was estimated at 10 dogs/km^2^, in this study which is lower compared to findings from some other parts of the world, such as 1081 dogs/km^2^ in San Francisco de Campeche, a city in Mexico [50], 242 dogs/km^2^ in Tiswadi taluka, in the state of Goa in southwest India [61], 52 dogs/km^2^ in Dhaka city, Bangladesh [48], 57 dogs/km^2^ in Aarey Milk Colony, a suburb of Mumbai, India [46], 119 dogs/km^2^ in the Coquimbo region of Chile [1], 334 dogs/km^2^ in Iringa, Tanzania [62], and 225 dogs/km^2^ in Shimotsui, Kurashiki city, Japan [63], but rather high compared to the 1.3 dogs/km^2^ reported by Rinzin [59] in Bhutan. However, our finding was nearly consistent and comparable with the 14 dogs/km^2^ and 12.25 dogs/km^2^ reported by Hossain et al. [40] in Raipura Upazila, Bangladesh and Bouaddi et al. [38] in EL Jadida, Morocco, respectively. Nevertheless, the dog density was reported to completely vary, from as low as 1 dog/km^2^ [1] and 1.3 dogs/km^2^ [59] to as high as 1380 dogs/km^2^ [1] and 2930 dogs/km^2^ [54]. Therefore, this result generally reflects the lower abundance of free-roaming dogs than that reported elsewhere, probably due to the low percentage of owned dogs and high number of confined owned dogs, or possibly due to the recent fencing of public places, such as parks, which has restricted the shelter of stray dogs and interference of humans in garbage sites, which has restricted accessibility of free-roaming dogs to foods, consequently causing them to move to semi-urban and rural areas.

The FRD-to-human ratio (1:315/3.17 dogs per 1000 people) obtained in this study is considered quite high compared to the 1:828 in Dhaka city, Bangladesh, and 1:493 in Punjab, Pakistan, which were estimated and reported by Tenzin et al. [48] and Shah, [63], but is considered low compared to (1:14), (1:14.3), (1:139), (1:4.7), (1:6.7), (1:2.3) and (1:120) that were estimated and reported by Gsell et al. [62], Chaudhari [64], Ayiedun and Olugasa [65], Kato et al. [54], Pimburage et al. [44], Cortez-Aguirre et al. [50]. and Hossain et al. [40] in Tanzania, the Haryana state of India, Ilorin in Nigeria, Kathmandu in Nepal, Sri Lanka, a city in southern Mexico and rural Bangladesh, respectively. Additionally, it is almost comparable with the 1:400 reported by and Özen et al. [49]. However, the density of the dog population may vary between different areas and countries, due to differences in sociocultural, environmental and dog-control strategies. A likely reason for the result obtained here (Herat city) may be the cultural and religious barriers to keeping dogs.

The highest number of FRDs was recorded in districts 5, 9 and 15 of Herat city. The possible factors associated with the highest abundance of FRDs in district 5 were the high density of the human population, unrestricted properties for the roaming of dogs (parks, parking lots), many commercial storerooms, large food and construction markets and the existence of many restaurants and dump sites that directly or indirectly provided food and shelter for FRD populations. Additionally to the factors mentioned above, a part of district 9 is the transportation point and vehicle station of several villages, and when travellers leave the area, they usually leave behind edible garbage along the roads and sidewalks, which attracts more FRDs in the area during times of lower human interference (early morning/evening/night). The density of the FRD population in district 15 was probably due to the high density of people, the size of the area, the presence of more owned dogs, the relative poverty of the residents, the lack of municipal services such as regular garbage collection and the bringing and releasing of unwanted dogs to this area from other regions by their owners (according to several residents).

The sex ratio of FRDs in this study skewed towards males (61%), which is consistent with the findings from other parts of the world [7,21,40,41,44,48,50,51], but is in contrast to (55.3% females) reports in Bali and Indonesia by Hiby et al. [33]. However, many studies have reported male dogs outnumbering females in FRD populations, and this disparity could be attributed to the preference of communities for male dogs or a high female dog mortality rate [21]. Based on our visual observation, 7.61% of dogs were identified as pregnant in this study, which is close to the percentage reported by Rinzin et al. [59] of 6.2%.

The age distribution was highly skewed toward adults, which is similar to the findings reported by others [7,21,40,50,59]. Possibly, the low density of puppies and young FRDs is due to the death of FRD puppies by accidents and deliberate poisoning or the absence of any communal or group care of pups by other females, which would agree with the report by Tiwari et al. [21]. Furthermore, it may be due to their susceptibility to diseases, such as distemper and canine parvovirus, which has resulted in high mortality and a consequently lower percentage of young dogs.

A high proportion of dogs (60.13%) were found to have a good body-condition score (BCS3), which indicates higher survivability and better capabilities to reproduce. Our finding is quite similar to what was recorded by other researchers, but is slightly lower than the finding by Cortez-Aguirre et al. [50]. However, most of the identified FRDs in Herat city had an ideal body-condition score, which is probably related to the availability of food (edible garbage), feeding of FRDs by the community or the lower density of dogs, which causes less competition, or, perhaps, a proportion of these dogs were actually free-roaming owned dogs, freely allowed to roam the neighbourhood.

Generally, the health status of FRDs was good, considering their stray status. The percentage of FRDs with skin lesions or dermatitis which was recorded (mostly in the males) in this study was low compared to the findings reported by Tenzin et al. [48], Cortez-Aguirre et al. [50] and Hiby et al. [33] in Dhaka city in Bangladesh, Campeche city in Mexico and Sanur in Bali, respectively Moreover, the fact that skin lesions were recorded mostly in male dogs is consistent with the finding by Cortez-Aguirre et al. [50], but different from the finding by Tenzin et al. [48]. Emaciation was recorded only in 2.6% of the dogs in this study, which is inconsistent with the findings of 23.2% by Hiby et al. [33], 12% by Tiwari et al. [21] and 8.9–10.6% by Hiby and Hiby [14]. In this study, most dogs had no visible injuries, and only 10.02% of the FRD exhibited visible injuries or wounds, in which our finding concurs with the finding reported by Hiby et al. [33].

A high proportion of FRDs were seen on the roads and sidewalks, which is possibly correlated with the abundance of edible litter since garbage is not collected regularly, and this finding concurs with those reported by Kato [54], Tenzin et al. [48] and Cortez-Aguirre et al. [50]. It is believed that the distribution of roaming dogs is associated with the management of garbage-collection points [48]. In this study, around 48% of the FRDs were found around a 20 m radius of garbage sites, which is quite similar to the finding of Tiwari et al. [21]. Similarly, a high percentage of dogs (68%) were active, which is comparable to that observed and reported by Tiwari et al. [21], with 43%.

In this study, around 68% of dogs were observed in groups, which was comparable with the 59.2% reported in Antananarivo, Madagascar [36]. The most common composition of social structure was male groups and male–female groups. This is probably due to the outnumbering of males compared to females, where the male groups incorporate females depending on their availability. However, there are significant dangers in forming groups, as groups of roaming dogs are not normally tolerated by the public.

## 5. Conclusions

In conclusion, having knowledge of the FRD population is crucial for the effective control of dog populations and the implementation of rabies-control programmes. This study provided the size of the population, demographic data, health status and dynamics of FRDs in Herat city, Afghanistan, by using the photographic sight–resight method. The human-to-FRD ratio was estimated at 315:1, with a density of 10 dogs/km^2^. A higher proportion of dogs were male and they were predominately adults. Generally, the health status of FRDs was good, with the majority of the FRD population found to have an ideal body-condition score. Skin lesions/dermatitis, lameness and emaciation were recorded only in 18% of FRDs. Most dogs were seen on the roads and sidewalks and a high proportion were active, and almost half of them were seen near garbage sites. Nearly half of them did not tolerate human presence and fled whenever approached. The abundance of FRDs and lack of prophylactic programmes targeting FRDs have generated public health problems in Herat city. The baseline information gained by this study could be used by the local government for the planning, implementation and monitoring of effective dog-population-control interventions, as well as for animal health planning, such as in canine-rabies control. Therefore, we feel there is a value in further studies at different times of the year. Additionally, similar studies are recommended in rural areas of the Herat province to obtain comprehensive FRD ecological data for the effective control of the dog population and rabies.

## Figures and Tables

**Figure 1 animals-13-01126-f001:**
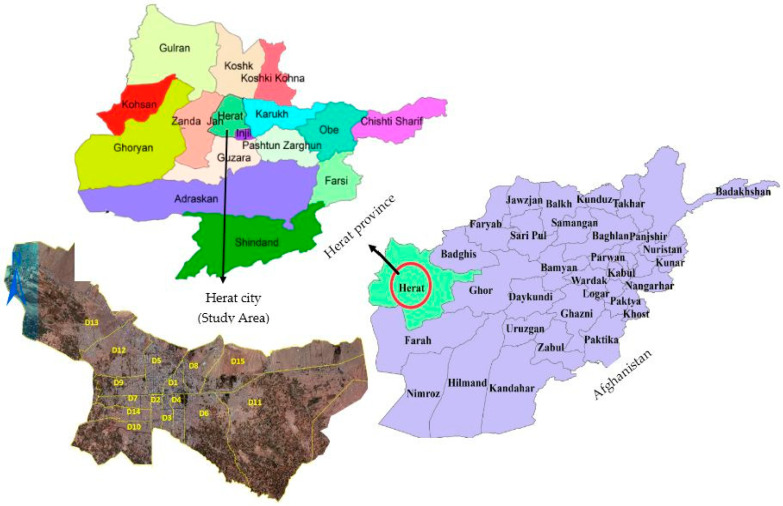
Map of Herat city (study area) and its urban districts. (D = district).

**Figure 2 animals-13-01126-f002:**
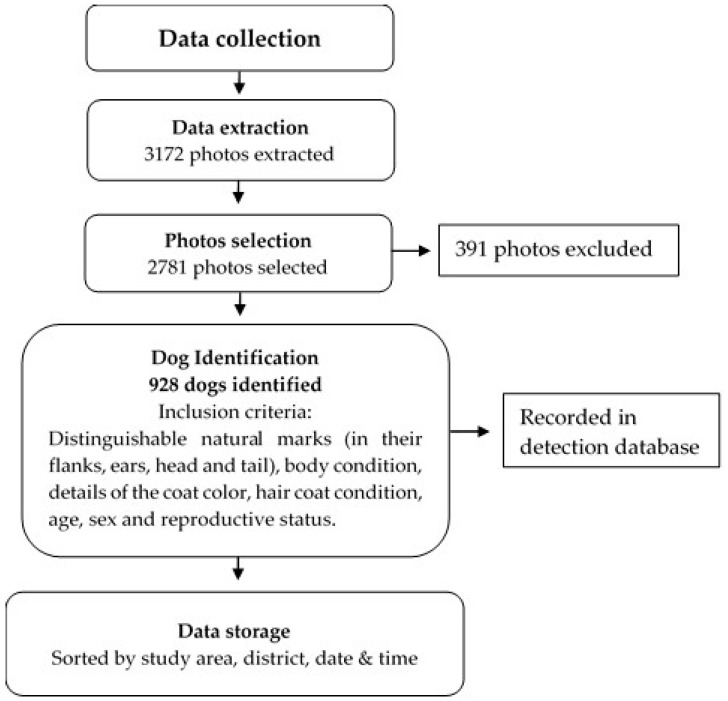
Flowchart of the data collection through field survey using PSR method. 391 photos were excluded due to poor quality and duplication.

**Figure 3 animals-13-01126-f003:**
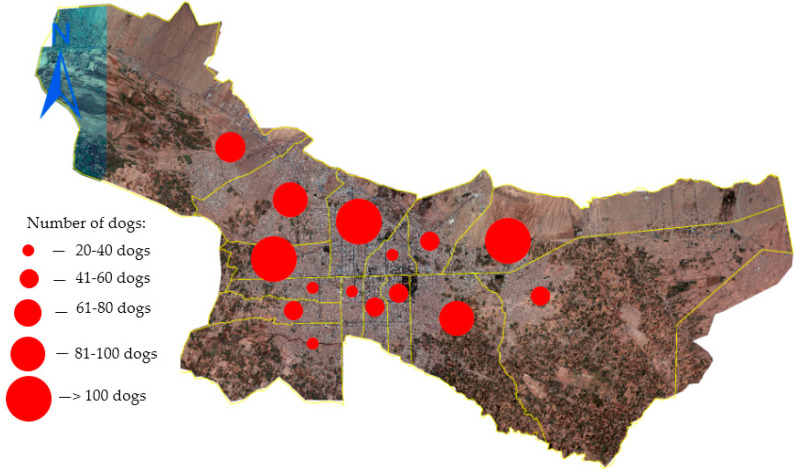
Total number of identified FRDs in 15 districts of Herat city during the survey period.

**Figure 4 animals-13-01126-f004:**
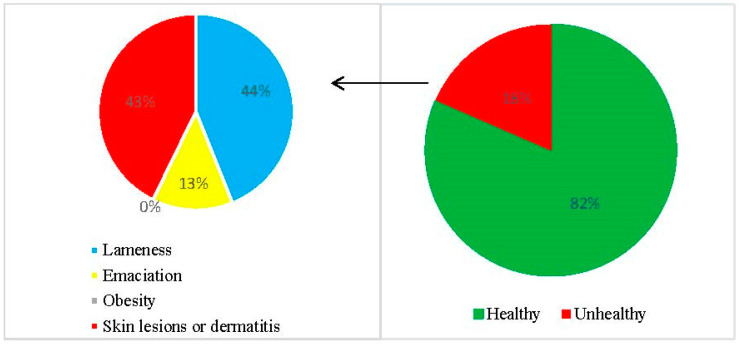
Health status of FRDs during the survey based on visual observation (direct observation during the survey or from reviewing photographs).

**Figure 5 animals-13-01126-f005:**
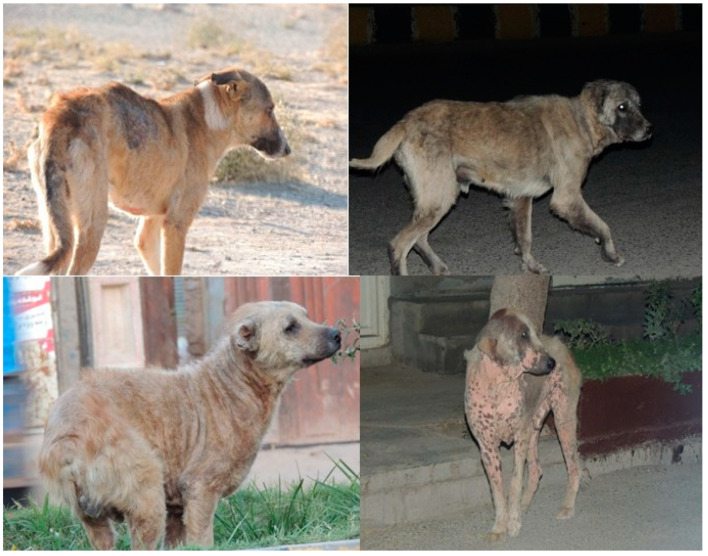
Selected images of dogs surveyed. Evidence of emaciation and skin lesion or dermatitis in FRDs recorded in Herat city.

**Figure 6 animals-13-01126-f006:**
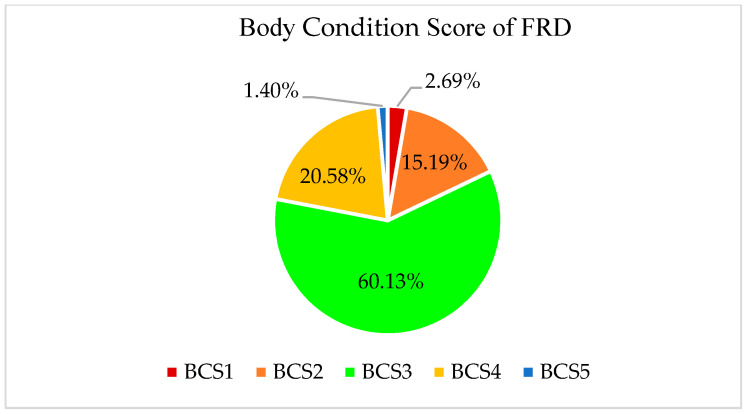
Body-condition score (BCS) of FRDs (assessed based on visual observation with regards to features that the ACIM organization has defined for each category of body-condition scoring of dogs).

**Figure 7 animals-13-01126-f007:**
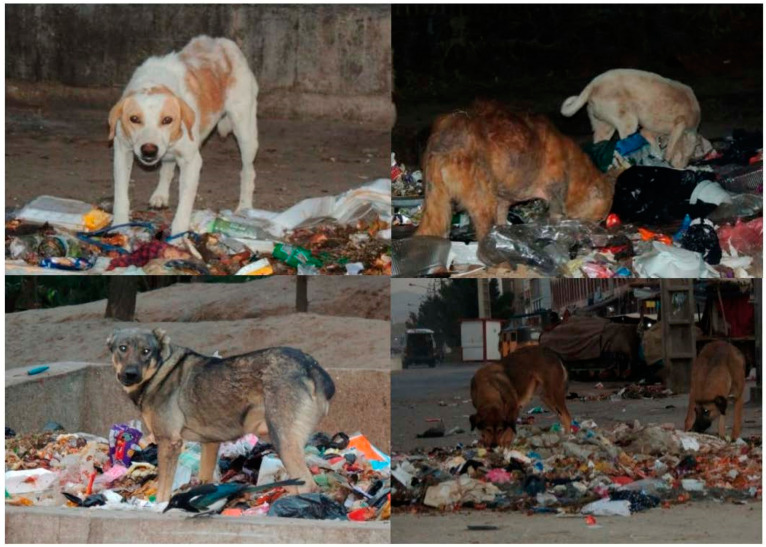
Images of FRDs in Herat city foraging edible materials from uncollected garbage along the streets and sidewalks.

**Figure 8 animals-13-01126-f008:**
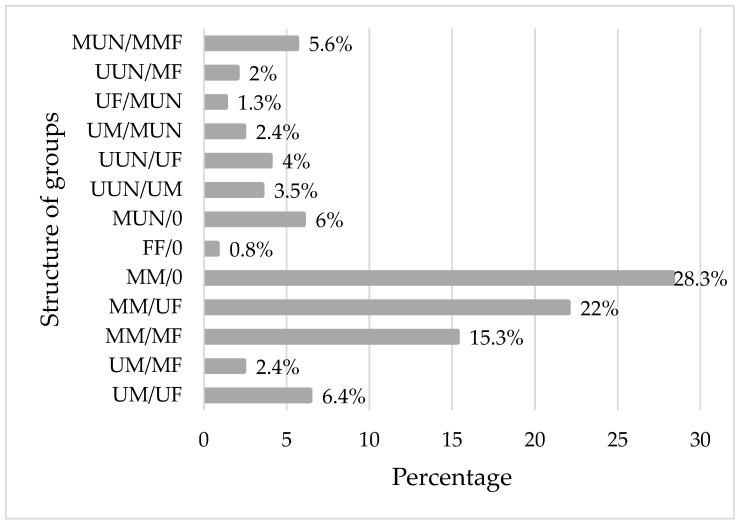
Group structure of FRD population during the survey period in Herat city. UM/UF (one male with one female), UM/MF (one male with multiple females), MM/MF (multiple males with multiple females), MM/UF (multiple males with one female), MM/0 (group of multiple males), FF/0 (group of multiple females), MUN/0 (group of multiple unverified gender), UUN/UM (one of unverified gender with one male), UUN/UF (one of unverified gender with one female), UM/MUN (one male with multiple of unverified gender), UF/MUN (one female with multiple of unverified gender), UUN/MF (one of unverified gender with multiple females), MUN/MMF (multiple of unverified gender with multiple males and females).

**Figure 9 animals-13-01126-f009:**
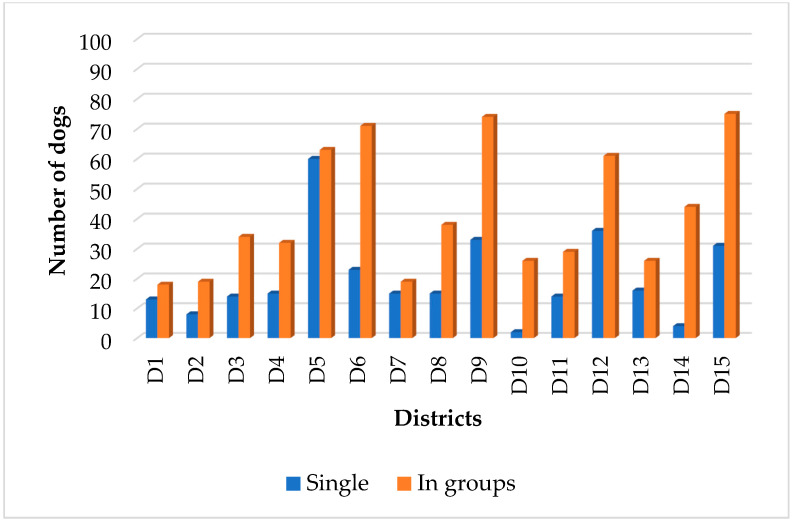
Social behaviour/structure of FRD population during the survey period in each district of Herat city.

**Table 1 animals-13-01126-t001:** Demographic details (sex, age distribution and reproductive status) of FRDs observed in 15 districts of Herat city.

Districts	Total Dogs	SEX			Age Category			Reproductive Status			
		Male (%)	Female (%)	Nonverifiable (%)	Puppy (%)	Young (%)	Adult (%)	Pregnant (%)	Lactating (%)	Female in Oestrus (%)	Nonverifiable (%)
1st	31	31 (100)	0 (0)	0 (0)	1 (3)	3 (10)	27 (87)	0 (0)	0 (0)	0 (0)	0 (0)
2nd	27	17 (63)	5 (18.5)	5 (18.5)	0 (0)	2 (7)	25 (93)	1 (20)	1 (20)	0 (0)	3 (60)
3rd	48	29 (60.5)	13 (27)	6 (12.5)	1 (2)	9 (19)	38 (79)	0 (0)	2 (15)	0 (0)	11 (85)
4th	47	31 (66)	9 (19)	7 (15)	5 (11)	6 (13)	36 (76)	0 (0)	1 (11)	2 (22)	6 (67)
5th	123	73 (59)	22 (18)	28 (23)	3 (2)	23 (19)	97 (79)	4 (18)	1 (4)	3 (14)	14 (64)
6th	94	62 (66)	19 (20)	13 (14)	6 (6)	2 (2)	86 (92)	2 (10.5)	2 (10.5)	2 (10.5)	13 (68.5)
7th	34	22 (65)	9 (26)	3 (9)	0 (0)	3 (9)	31 (91)	1 (11)	1 (11)	0 (0)	7 (78)
8th	53	24 (45)	18 (34)	11 (21)	7 (13)	2 (4)	44 (83)	1 (5.5)	1(5.5)	1 (5.5)	15 (83.5)
9th	107	52 (48.5)	18 (17)	37 (34.5)	4 (4)	14 (13)	89 (83)	0 (0)	4 (22)	1 (6)	13 (72)
10th	28	17 (61)	8 (28)	3 (11)	1 (3.5)	1 (3.5)	26 (93)	0 (0)	1 (12)	0 (0)	7 (88)
11th	43	30 (70)	8 (19)	5 (11)	1 (2)	0 (0)	42 (98)	1 (12)	0 (0)	0 (0)	7 (88)
12th	97	51 (52)	25 (26)	21 (22)	6 (6)	14 (15)	77 (79)	3 (12)	1 (4)	1(4)	20 (80)
13th	42	27 (64)	10 (24)	5 (12)	1 (2)	4 (10)	37 (88)	1 (10)	1 (10)	1 (10)	7 (70)
14th	48	35 (73)	6 (12.5)	7 (14.5)	2 (4)	5 (11)	41 (84)	1 (17)	0 (0)	1 (17)	4 (66)
15th	106	61 (58)	27 (25)	18 (17)	1 (1)	8 (8)	97 (91)	0 (0)	2 (7)	5 (19)	20 (74)
Total	928	562 (61)	197 (21)	169 (18)	39 (4.20)	96 (10.34)	793 (85.45)	15 (7.61)	18 (9.13)	17 (8.62)	147 (74.62)
Chi-Square and *p*-value		χ2 = 65.33, df = 28, *p* < 0.001			χ2 = 57.63, df = 28, *p* < 0.001			χ2 = 51.81, df = 39, *p* < 0.08			

The age of FRDs were estimated, based on visual observation, as puppy (<6 months), young (6 months to 1 year) and adult (>1 year). The percentages of sex and age categories in each district were calculated based on the total dogs in each district, and the reproductive status was calculated from the total female dogs in each district.

**Table 2 animals-13-01126-t002:** Details of locations FRDs were sighted during the survey in each district of Herat city.

Location during the Survey												
Districts	Road (%)	Sidewalk (%)	Park (%)	Playground (%)	Field (%)	Parking Lot (%)	Underneath a Car (%)	Bush (%)	Dump Sites= (%)	Slaughterhouse (%)	Biological-Waste-Disposal Site (%)	Boulevard (%)
1st	1 (3)	25 (81)	0 (0)	0 (0)	0 (0)	5 (16)	0 (0)	0 (0)	0 (0)	0 (0)	0 (0)	0 (0)
2nd	17 (63)	3 (11)	0 (0)	0 (0)	0 (0)	1 (4)	0 (0)	0 (0)	1 (4)	0 (0)	0 (0)	5 (18)
3rd	24 (50)	13 (27)	0 (0)	0 (0)	0 (0)	0 (0)	0 (0)	0 (0)	2 (4)	0 (0)	0 (0)	9 (19)
4th	17 (36)	29 (62)	0 (0)	0 (0)	0 (0)	0 (0)	0 (0)	0 (0)	0 (0)	0 (0)	0 (0)	1 (2)
5th	53 (43)	39 (32)	4 (3)	1 (1)	2 (1)	0 (0)	0 (0)	0 (0)	2 (2)	0 (0)	0 (0)	22 (18)
6th	35 (37)	30 (32)	0 (0)	0 (0)	2 (2)	0 (0)	0 (0)	0 (0)	26 (28)	0 (0)	0 (0)	1 (1)
7th	22 (65)	8 (23)	1 (3)	0 (0)	0 (0)	0 (0)	0 (0)	1 (3)	0 (0)	0 (0)	0 (0)	2 (6)
8th	34 (64)	16 (30)	0 (0)	0 (0)	0 (0)	0 (0)	0 (0)	0 (0)	3 (6)	0 (0)	0 (0)	0 (0)
9th	33 (31)	38 (35)	8 (7)	0 (0)	17 (16)	6 (6)	0 (0)	0 (0)	1 (1)	0 (0)	0 (0)	4 (4)
10th	12 (43)	3 (11)	0 (0)	0 (0)	0 (0)	0 (0)	0 (0)	0 (0)	6 (21)	0 (0)	0 (0)	7 (25)
11th	27 (63)	6 (14)	0 (0)	0 (0)	0 (0)	0 (0)	0 (0)	0 (0)	10 (23)	0 (0)	0 (0)	0 (0)
12th	27 (28)	46 (48)	0 (0)	1 (1)	6 (6)	12 (12)	2 (2)	1 (1)	0 (0)	0 (0)	0 (0)	2 (2)
13th	9 (21)	8 (19)	0 (0)	0 (0)	5 (12)	0 (0)	0 (0)	0 (0)	0 (0)	20 (48)	0 (0)	0 (0)
14th	19 (39.5)	18 (37.5)	0 (0)	0 (0)	9 (19)	0 (0)	0 (0)	0 (0)	0 (0)	2 (4)	0 (0)	0 (0)
15th	61 (57)	22 (21)	0 (0)	0 (0)	15 (14)	0 (0)	0 (0)	0 (0)	7 (7)	0 (0)	1 (1)	0 (0)
Total	391 (42.13)	304 (32.76)	13 (1.40)	2 (0.22)	56 (6.03)	24 (2.59)	2 (0.22)	2 (0.22)	58 (6.25)	22 (2.37)	1 (0.10)	53 (5.71)
Chi-Square and *p*-value			χ2 = 957.7, df = 154, *p* < 0.0001									

**Table 3 animals-13-01126-t003:** Details of reactions of sighted FRDs during the survey period in each district of Herat city.

Reaction during the Survey						
Districts	Approaches (Friendly) (%)	Approaches (Slowly, Low Posture) (%)	Growls/Barks (%)	Attacks the Surveyor (%)	Neutral (e.g., Ignores the Surveyor) (%)	Runs Away (%)
1st	1 (3)	12 (39)	2 (6)	0 (0)	9 (29)	7 (23)
2nd	0 (0)	4 (15)	0 (0)	1 (4)	13 (48)	9 (33)
3rd	0 (0)	12 (25)	4 (8)	1 (2)	12 (25)	19 (40)
4th	0 (0)	0 (0)	2 (4)	1 (2)	17 (36)	27 (58)
5th	1 (1)	3 (2.4)	4 (3.2)	1 (1)	52 (42)	62 (50.4)
6th	2 (2)	31 (33)	3 (3)	1 (1)	11 (12)	46 (49)
7th	1 (3)	1 (3)	1 (3)	0 (0)	18 (53)	13 (38)
8th	0 (0)	19 (36)	1 (2)	2 (4)	8 (15)	23 (43)
9th	0 (0)	3 (3)	6 (6)	0	55 (51)	43 (40)
10th	0 (0)	13 (46)	1 (4)	0 (0)	1 (4)	13 (46)
11th	0 (0)	17 (40)	2 (5)	0 (0)	1 (2)	23 (53)
12th	0 (0)	3 (3)	7 (7)	0 (0)	49 (51)	38 (39)
13th	0 (0)	2 (5)	4 (9.5)	0 (0)	16 (38)	20 (47.5)
14th	1 (2)	11 (23)	4 (8)	1 (2)	13 (27)	18 (38)
15th	1 (1)	45 (42)	5 (5)	2 (2)	8 (8)	45 (42)
Total	7 (0.75)	176 (18.97)	46 (4.96)	10 (1.07)	283 (30.5)	406 (43.75)
Chi-Square and *p*-value		χ2 = 278.9, df = 70, *p* < 0.0001				

**Table 4 animals-13-01126-t004:** Details of activity, social behaviour and sightings within 20 m of garbage points for FRDs sighted across fifteen districts of Herat city.

Districts	Activity			Social Behaviour		Proximity to Garbage Point	
	Active (%)	Inactive (%)	Feeding/Foraging (%)	Single (%)	In Packs (%)	≤20 m (%)	>20 m (20)
1st	15 (48)	14 (45)	2 (7)	13 (42)	18 (48)	5 (16)	26 (84)
2nd	14 (52)	7 (26)	6 (22)	8 (30)	19 (70)	20 (74)	7 (26)
3rd	25 (52)	16 (33)	7 (15)	14 (29)	34 (71)	24 (50)	24 (50)
4th	25 (53)	21 (45)	1 (2)	15 (32)	32 (68)	27 (57)	20 (43)
5th	76 (62)	27 (22)	20 (16)	60 (49)	63 (51)	35 (28)	88 (71)
6th	60 (64)	25 (26.5)	9 (9.5)	23 (24)	71 (76)	66 (70)	28 (30)
7th	20 (59)	8 (23.5)	6 (17.5)	15 (44)	19 (56)	21 (62)	13 (38)
8th	25 (47)	20 (38)	8 (15)	15 (28)	38 (72)	37 (70)	16 (30)
9th	61 (57)	35 (33)	11 (10)	33 (31)	74 (69)	25 (23)	82 (77)
10th	15 (54)	10 (36)	3 (10)	2 (7)	26 (93)	18 (64)	10 (34)
11th	24 (56)	15 (35)	4 (9)	14 (33)	29 (67)	31 (72)	12 (28)
12th	48 (50)	40 (41)	9 (9)	36 (37)	61 (63)	37 (38)	60 (62)
13th	25 (60)	14 (33)	3 (7)	16 (38)	26 (62)	13 (31)	29 (69)
14th	29 (60)	11 (23)	8 (17)	4 (8)	44 (92)	33 (69)	15 (31)
15th	67 (63)	34 (32)	5 (5)	31 (29)	75 (71)	49 (46)	57 (54)
Total	529 (57)	297 (32)	102 (11)	299 (32)	629 (68)	441 (48)	487 (52)
Chi-square and *p* value		χ2 = 38.70, df = 28, *p* < 0.08		χ2 = 45.10, df = 14, *p* < 0.0001		χ2 = 127.9, df = 14, *p* < 0.0001	

## Data Availability

Data sharing is not applicable to this article.

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
