# Peer review of "Evaluation of Dynamics, Demography and Estimation of Free-Roaming Dog Population in Herat City, Afghanistan"

_animals, 2023, doi:10.3390/ani13071126_

Round 1
Reviewer 1 Report (Previous Reviewer 2)

Author Response
.

Reviewer 2 Report (New Reviewer)
Dear Authors,
The paper entitled "Evaluation of Dynamics, Demography and Estimation of Freeroaming Dog Population in Herat City, Afghanistan"aims to estimate the FRD population and evaluate its demographic characteristics in Herat city.
The chosen research topic is very important because of the uncontrolled spread of rabies disease in human populated areas and the possibilities for its prevention and control.
The results of the study may be useful for animal health planning to develop effective dog population control programs and for planning national rabies prevention programs.
Abstracts:
The abstract should be revised according to the journal's guidelines, as information on the research methods used is missing.
Keywords:
Should consist of only 5 keywords, no more. Please correct according to the journal's guidelines.
After reading the paper, I have comments and suggestions for improving the paper, which are as follows:
1. Introduction
This chapter is well developed and introduces the research topic. However, the following was missing at the end of the chapter: the purpose of the study, the research questions and possibly the research hypothesis.
2. Materials and methods
This chapter is well developed.
I suggest explaining why exactly the Lincoln-Petersen formula with Chapman correction was adopted for the study? Has anyone already used this method for similar research?
Technical errors
Figure 2. diagram of data collection through field survey by PSR method - needs improvement as it is illegible. 3.
3. Results
Overwhelmingly, the results are well and interestingly presented and described with a table and figure. The results are presented and described in a good way, they are very interesting
4 Location of free-roaming dogs at the time of the survey
There is a lot of information in this subsection in the form of text and figures. Therefore, the reception for the reader may be unreadable. I suggest not to repeat the values in the tables and only give references.
5. Conclusions
Recommendations were missing
Technical errors that should be removed:
Figure lacks source
I suggest improving the literature according to the rules of the journal
All in all, I recommend the paper for publication in the journal Animals with minor changes.
Kind regards,
Reviewer
Author Response
.

Reviewer 3 Report (New Reviewer)
This is an interesting manuscript providing a lot of valuable information about the FRD population of this city. The analyses tend to be done by district (although not all variables are analyzed this way). However, there isn’t really any information about the districts that might indicate why fewer or more dogs were seen there, or why, for example so few dogs were in the road in the 1st district and so many in several others including the 11th. If district is an important variable to study, the authors need to explain why and include some general descriptions. Paragraph starting on line 356: this and the following paragraphs are really important information. Some of these suggested reasons are actually characteristics that could be analyzed (human population density, numbers of dump sites, size of the district, income or poverty level). If these are potential predictors, the authors should consider an additional manuscript with a more sophisticated analysis to explore and see if these hypothesized factors are actually important and suggest this as a future research direction. Additionally, the authors could document them as important in other studies. This would be more useful than a listing of human to dog ratios from around the world.
Line 70: culling is not an effective way to control rabies in dogs.
Section 2.2: What was the weather like during this study period? Did rain or bad weather change activities or when blocks were driven? Is this study period during the dog breeding season or not? How were the blocks determined? Are they related to the urban districts in the figure? Was each block’s route done on more than on the one day (morning and evening)? Please add to the text. Ideally, the map of the driving routes would also be provided in the manuscript.
Section 2.3: How were the “predetermined streets and alleys” decided upon? How much of each of the blocks are covered by these routes? Please add to the text.
Line 131: how was age of young dogs determined? What characteristics were used? Please add to text.
Section 2.4: who looked at the photos? How many different people? How were they trained so that the identification was accurate? Why were photos excluded (figure 2)? Please add to the text here (move from results).
Section 2.5: there is a regression in the results, what software was used for that and what are the x and y variables? And clarify that the demographic data were compared by districts (and any other comparisons that have been done). Please add.
Section 3.1: please add the actual data for the variables in the resight equation. And the variables used for the x and y in the regression on line 190.
Line 187-9: the part in parentheses should be in the discussion instead of here.
Line 196 and 198, 360: please replace gender with sex here as well.
Line 218: “macroscopic lesions” should instead be skin lesions or dermatitis as used elsewhere and defined in the methods.
Figure 4 is unnecessary.
Section 3.3 and throughout: please round percentages to whole numbers or 0.5 as was done in the first table.
Section 3.3: Body condition score and health information: I would like to see this included in one table with the other demographic characteristics in the first paragraph of section 3.2. Include the confidence intervals. That will make reviewing these data much easier for the reader. And then figure 6 is unnecessary.
Table 2 and 3: can these locations be ordered from most to least common (by total dogs)? Are these row percentages I assume? Please include in the table for clarity.
Line 262 and others: were the behaviors of the dogs (3.4) associated with being in groups or alone (3.5)? That seems important to consider with sex and age.
Line 276: add to this last sentence: across the districts.
Figure 8: please order from most to least common (or vice versa).
Discussion: Overall the discussion is more of a review of the results without indicating why the findings might or might not be useful for animal health planning and in managing the population of dogs. Or for rabies control programs. I’d like to see the authors focus more on somewhat similar studies from the literature and include how the data were used for animal health, population control or rabies control planning or implementation instead of just listing human:dog ratios or restating results. Additionally, based on the overall findings, what specific recommendations do the authors feel are important for this population in this city? Are there examples of population or rabies control already published to apply here? Please consider how to make the discussion more about how and why this study should be used and to prioritize actions to take with this population given the relatively low numbers, good health, and adult status. Perhaps include in the conclusion, which districts ought to be prioritized for specific programs?
Line 329-30: please reference these statements about dog ownership and provide additional information about the parks being fenced (only in specific locations? Where fewer dogs were found?).
Line 345: please add some additional information about barriers to keeping dogs for context for readers not familiar with the culture or location.
Line 375: was body condition score related to age group or sex or skin lesions? That could be valuable information to include in the manuscript.
Line 394: I would also expect that if the observers were in a car, roads and sidewalks would be the most visible areas, wouldn’t they? Please edit for clarity.
Author Response
.

This manuscript is a resubmission of an earlier submission. The following is a list of the peer review reports and author responses from that submission.
Round 1
Reviewer 1 Report
This paper tackles an important issue namely the knowledge of the structure of free roaming dog population. This is an issue of multiple interests including those developed in this paper.
While it is important to control the population of FR dogs, it is also important to understand the social structure of free roaming dogs for different applied reasons.
The rationale of this paper is not clearly exposed in particular because of the high number of repetitions, which is an unfortunate characteristic of this paper.
While from one table, we could understand that most of the dogs are living in groups, it would have be nice to know the compostion of the group a very important and overlooked data about dog social structures. What is the composition of the different groups MM/MF or UM/MF, etc. ? It would have be nice to get this information along [or even instead !] the lenghty tedious tables to read.
The sampling methods are not clearly explained. One understand that the observations are made from a car, so some sentences in the results are awkward and do not reflect the reactions of dogs to humans.
This paper needs to be focused on one point. The data could be used to publish several other papers about the life and especially the social life of FR dogs. The authors should be thanked for their collection of such data
The observations contained in this paper are important, but the scope is quite narrow and it is difficult to understand the importance of this paper in a general scientific journal as Animals. Again data about the social structures of these FRDs would have been more suitable for Animals.
This paper should be severely shortened to be more effective.
Some additional comments
L 119 Section 2.2 What kind of sampling is completed ? This section is far from being explicit and should be developed.
L39 Canis instead of Canine
Line 41-42 / Line 51-52-53 Useless. Off the focus of the paper.
Line 93 Elsewhere or worldwide ?
Line 94-96 Repetition. Delete to shorten
L 135 What is a convenient distance.
L 139 Second part of the sentence useless. Delete
L 168 there is one word missing : « n1 is the of dogs » !
L 199 Figure 3. A map with circles of different sizes related to the number of dogs in a given district might be more informative than a number of 1 to 15.
Line 205. The reading of the paper is made difficult by setting parentheses in the middle of the sentences. As an example, I suggest to read line 205 as « 60.6% were males (562/928; 95% CI = 65.6% - 57.6%) »
L 210 Delete « in FRD »
L 218 The caption of the table is not enough detailed. The reader has to guess – and he should not guess!- that number within parentheses is the percentage relative to the total number of dogs within a district.
L 282 This figure is useless. Replace by clear sentences instead.
L 274 Rewrite this sentence
L 335-339 Difficult to understand high and low ratio. How 315:1 could be more than 828:1. The reverse would be right 1:315 is more than 1:828
L 345-350. It would have been informative to cite the countries along the densities.
Author Response
Thank you so much for taking the time to comment on our paper. Your comments have very important value and help for modifying and perfecting our papers, and they also have certain guiding significance for our future research. We have revised the paper based on your comments.

Reviewer 2 Report
Line 10 – “existing”
Line 16 – Kabul has implemented a dog rabies vaccination program for several years now – should be mentioned.
1,821 dogs Human:dog ratio of 315:1 Dogs per 1,000 is 3.17.
Male:female was 2.85:1
Herat area – 182 Km2
Majority healthy with ideal body score
Line 40 – correct the English (“most frequently owned pet species in the world”)
Line 48 – stray dog management is not difficult but municipal authorities seldom support/implement effective programs.
Line 114 – Herat is not at all densely populated. There are only around 4 people per square kilometer. For a dense city, look at Dakha with over 20,000 people per sq kilometer.
Line 336 a human dog ratio of 10,000:25 (i.e., 400:1) is very close to the ratio seen in Herat. The Dhaka estimate was observed in a very dense urban city 23,000 people per square Km compared to Herat at around 4 person per square kilometer). The relative dog population (dogs/1,000 people) varies inversely with human density so one would expect a higher relative dog population in Herat. I would recommend the authors read the report of the dog population surveys in Haryana State (India - https://www.wellbeingintlstudiesrepository.org/strfapop/3/) to gain some detail on human and dog density relationship. The PhD thesis by S I Shah (ECOLOGY AND DEMOGRAPHY OF THE DOG POPULATION IN RAWALPINDI DISTRICT.
SYED ISRAR SHAH, Dept of Zoology, Arid Agriculture University Rawalpindi, Pakistan) would also be useful to cite since he did detailed assessments of dogs in Pakistan communities.
The ratio of dogs to people (dogs per 1,000 people is the measure I typically use) varies from around 3 in Herat to over 400 in some US states but tends to be somewhere between 100-250 in North America and Europe. It is lower in Islamic countries (around 10-40 in Turkey and 1-10 in Bangladesh and Pakistan).
The use of the measure dogs/square kilometer is not a useful measure. In cities, there are a lot more dogs per square kilometer than in rural areas but the number of dogs per 1,000 people is much higher in rural areas than in cities. And, in rural areas, the dogs will congregate around human homes (because that is where the food is) so they are not spread evenly across the countryside. I would strongly urge the authors to track dogs per 1,000 humans rather than dogs per sq kilometer.
Line 369 – Given the relatively low population of dogs in herat, I would expect the dogs to be clustered around food sources. But, again, if the authors were to plot the number of dogs per 1,000 people against the log of human density in each district of Herat, I would expect they would find an inverse relationship.
Line 382 – the survival of puppies in the FRD population is low (around 75% mortality). Once a dog has reached one year of age, it will live for several years at least.
Lines 406-413 – It is widely claimed that FRD are supported by garbage but the few studies that have actually looked at the issue (e.g. Morters et al, 2014 - https://besjournals.onlinelibrary.wiley.com/doi/full/10.1111/1365-2664.12279) report that there are too few calories in typical garbage to sustain the FRD population. In fact, FRD appear to be sustained by human feeding (not by scavenging garbage). I suspect the same is true in Herat 9as it is in Turkey) even given the antipathy towards dogs in Muslim countries.
There are very few data on dog populations and dog demographics in muslim countries so every published report from a muslim country will provide value. However, this could be even more valuable if the authors looked at relative dog populations (dogs per 1,000 people) in the fifteen districts of Herat and tracked those relative numbers against human density in each district.
Author Response
Thank you so much for taking the time to comment on our paper. These comments have very important value and help for modifying and perfecting our papers, and they also have certain guiding significance for our future research. We have revised the paper based on these comments.
Point 1: Kabul has implemented a dog rabies vaccination program for several years now – should be mentioned.
Response 1: Dear reviewer, thank you so much for your constructive comment. We have been mentioned the rabies vaccination program in Kabul city according to reviewer comment in revised manuscript The country has the highest number of human rabies cases in South Asia at 5.7 per 100,000 population [34]. For purpose of dog population control and rabies the country had been started to adopt mass dog vaccination with neutering program in Kabul and tens thousands of dogs were vaccinated against rabies between 2017-2019 [34] but the program has not covered all province of Afghanistan. However, abundance number of FRD, lack of regular vaccination, absence regular animal birth control program, limited human resources, and insufficient funds are the significant constraints to control of rabies [34].
Point 2: 1,821 dogs Human:dog ratio of 315:1 Dogs per 1,000 is 3.17.
Response 2: Dear reviewer, thank you so much for your constructive comment. We have been added “3.17 dogs per 1,000 populations “according to reviewer comment in revised manuscript in result section.
Point 3: Male:female was 2.85:1
Response 3: Thank you so much for your constructive comment, we have been modified according to reviewer comment in revised manuscript.
Point 4: Herat area – 182 Km2
Response 4: Thank you so much for your constructive comment, we have been modified according to reviewer comment in revised manuscript.
Point 5: Majority healthy with ideal body score
Response 5: Thank you so much for your constructive comment, we have been modified according to reviewer comment in revised manuscript. Majority healthy with ideal body score
Point 6: Line 40 – correct the English (“most frequently owned pet species in the world”)
Response 6: Thank you so much for your constructive comment, we have been modified according to reviewer comment in revised manuscript. It is the most frequently owned pet species in the world
Point 7: Line 48 – stray dog management is not difficult but municipal authorities seldom support/implement effective programs.
Response 7: Thank you so much for your constructive comment, we have been modified according to reviewer comment in revised manuscript. The management of stray dogs are a challenge and often involves capture, treatment, neutering and release, and still considered a significant problem in developing and developed countries
Point 8: Line 114 – Herat is not at all densely populated. There are only around 4 people per square kilometer. For a dense city, look at Dakha with over 20,000 people per sq kilometer.
Response 8: Thank you so much for your comment. You are completely right, Hera city is the densest, and most populated city compered to other city of Afghanistan and being the second populated city after Kabul.
Point 9: Line 336 a human dog ratio of 10,000:25 (i.e., 400:1) is very close to the ratio seen in Herat. The Dhaka estimate was observed in a very dense urban city 23,000 people per square Km compared to Herat at around 4 person per square kilometer). The relative dog population (dogs/1,000 people) varies inversely with human density so one would expect a higher relative dog population in Herat. I would recommend the authors read the report of the dog population surveys in Haryana State (India - https://www.wellbeingintlstudiesrepository.org/strfapop/3/) to gain some detail on human and dog density relationship. The PhD thesis by S I Shah (ECOLOGY AND DEMOGRAPHY OF THE DOG POPULATION IN RAWALPINDI DISTRICT.
SYED ISRAR SHAH, Dept of Zoology, Arid Agriculture University Rawalpindi, Pakistan) would also be useful to cite since he did detailed assessments of dogs in Pakistan communities.
Response 9: Dear reviewer, thank you so much for your comment. We have considered above suggested article and thesis in the revised manuscript and cited in discussion section.
Point 10: The use of the measure dogs/square kilometer is not a useful measure. In cities, there are a lot more dogs per square kilometer than in rural areas but the number of dogs per 1,000 people is much higher in rural areas than in cities. And, in rural areas, the dogs will congregate around human homes (because that is where the food is) so they are not spread evenly across the countryside. I would strongly urge the authors to track dogs per 1,000 humans rather than dogs per sq kilometer.
Response 10: Dear reviewer, thank you so much for your constructive point. We have considered above suggestion in the revised manuscript and we have been added the number of dogs pre 1,000 people. Also we calculated the number dogs per km2. Because we believe the number dogs per km2 determines how many dogs a resident will meet on their way to school, work, etc.
Point 11: Line 369 – Given the relatively low population of dogs in herat, I would expect the dogs to be clustered around food sources. But, again, if the authors were to plot the number of dogs per 1,000 people against the log of human density in each district of Herat, I would expect they would find an inverse relationship.
Response 11: Dear reviewer, thank you so much for your comment. Unfortunately, the human population in each district is not available and during the study period we referred many times to Municipal of Herat city and districts, but the such as information has not available.
Point 12: Lines 406-413 – It is widely claimed that FRD are supported by garbage but the few studies that have actually looked at the issue (e.g. Morters et al, 2014 - https://besjournals.onlinelibrary.wiley.com/doi/full/10.1111/1365-2664.12279) report that there are too few calories in typical garbage to sustain the FRD population. In fact, FRD appear to be sustained by human feeding (not by scavenging garbage). I suspect the same is true in Herat 9as it is in Turkey) even given the antipathy towards dogs in Muslim countries.
Response 12: Dear reviewer, thank you so much for your comment. Actually around half of FRD were seen and recorded in the garbage/or near to the garbage points during the survey period.
Point 13: There are very few data on dog populations and dog demographics in muslim countries so every published report from a muslim country will provide value. However, this could be even more valuable if the authors looked at relative dog populations (dogs per 1,000 people) in the fifteen districts of Herat and tracked those relative numbers against human density in each district.
Response 13: Dear reviewer, thank you so much for your positive comments. Unfortunately, the human population in each district is not available and during the study period we referred many times to Municipal of Herat city and districts, but the such as information has not available.
Round 2
Reviewer 1 Report
The authors have made a significant effort to answer to the reviewers. However some important points have not been understood right, arguably due to an unclear request from the reviewer. Particularly the Point 2 about the social structure of the packs was not understood. I'll try to make a clear request. A population in a large area is made of a certain proportion of males and females. Referring to the figure 3, there are several populations in the total area, their size differing from one district to another. This is of course an important information. Though dogs may form social groups or packs. The information that I consider of paramount interest for the issue of population structure and spread of diseases is the number of alone individuals vs the number of packs ( and the total number of dogs seen in packs)structure of the packs. In addition it is very important to know the social structure of the packs in each district. The classification of the social structure according to L. Fedigan 1982 might be as follows : UM/UF one male/one female groups, UM/MF one male multi females groups, MM/MF multi males / multi females groups and possibly other combinations as MM/0F groups only male groups, 0M/ MF groups of UF/MM groups one female / multi male groups. All these possible social structure are important in relation to reproduction and mating systems and so to the way diseases may spread out. In addition this information about social structure in dogs would constitute a very significant contribution to the understanding of dog biology. If the authors may include this information in their paper, its scientific value will be significantly increased.
Author Response
.

Round 3
Reviewer 1 Report
The efforts made by the authors to improve their paper is very much appreciated. The addition of social structure of groups of dogs represents a significant improvement of the paper. These data are invaluable as they are very seldom provided while the social structure of the dog is an overlooked issue. So the authors should be congratulated to provide such information. They should be regularly cited in later papers on dogs' social behavior.
Besides this significant improvement, it remains, some issues that should be corrected prior to publication. These points are listed below
The style of the paper might be simplified to avoid repeated redundance or useless wording
l 174 (Figure 2) instead of « as shown in figure 2 ».
l 195-196 Delete and add (Figure 3) after « survey period » line 189
Table 1 Some chi2 are displayed though these tests are either not explained in the text nor described in methods. So these results should be incorporated in the text. For example line 206 « Male dogs are signicantly overall more represented than other categories (chi2 =65,33, df = ?, p<0.001) ». These should be done for all performed chi squares. As shown in my suggestion degrees of freedom should be provided for each test. This applies to all tests performed in the different tables. Another solution is to delete all the tests in the tables as they are not used – unfortunately for some of them- in the text and so are useless. In the table 1, I suggest to change « Age group » into « Age category ».
line 229-230 replace the content within the parentheses by « (Figure 4 and Figure 5). Same for line 246 only « (Figure 6) to avoid unnecessary wording.
Line 263-264 Delete and add (Table 2) at the end of the sentence Line 256
Table 2 what is the purpose of preforming chi2 on the data of this table. See suggestion above. In adition if Chi2 are maintained, the degrees of freedom should be provided.
L 274-275 replace the last sentence with « (Table 3) ».
Idem line 282 an dlines 288-290.(Table 4 and Figure 7 respectively)
Table 4 very interesting though again chi2 tests are perfored though not used in the text. As an exmple, the authors may have written « dogs are significantly more often seen in groups that alone (chi2= xx, df= xx, p<x.xx) »
In addition there is an intersting result in that table that has been overlooked.While in most districts there are more groups than single individuals, in some districts the single sightings are as frequent as groups if binmail tests are to be performed. This should be commented along with the Figure 9 whiwh is excellent. In my opinion, the Figure 9 should renumberd as Figure 8 and follow direclty table 4. The figure 8 should be renumbered Figure 9 (comments beliw)
L 285-296 This sentence is not understandable. What is it meant ? If there are no differences how the Chi2 could be lesser than .95 ? Please correct and give df.
L 298-300 Simplfy as repeatitively suggested above
Actually the figure 8 should be commented in the text in details ( and include the reference to Figure 8). A suggestion « the most comm social structure of the groups is MM/0F, MM/UF and MM/MF. As already reported males are constituting social groups and as males outnumber females, the male groups incorporate femals depending on their availability, from only male groups to multimale groups.
This result souhld be discussed in the discussion section as theyr are VERY ORIGINAL.
So the discussion should be rewritten to incorporate the new IMPORTANT results.
Author Response
.
